# Antibiotics Usage and Resistance among Patients with Severe Acute Respiratory Syndrome Coronavirus 2 in the Intensive Care Unit in Makkah, Saudi Arabia

**DOI:** 10.3390/vaccines10122148

**Published:** 2022-12-14

**Authors:** Ahmed Kabrah, Fayez Bahwerth, Saad Alghamdi, Alaa Alkhotani, Ahmed Alahmadi, Mashari Alhuzali, Ibrahim Aljerary, Anwar Alsulami

**Affiliations:** 1Laboratory Medicine Department, Faculty of Applied Medical Sciences, Umm Al-Qura University, Makkah 21955, Saudi Arabia; 2Molecular Genetics Department, King Faisal Hospital, Ministry of Health, Makkah 21955, Saudi Arabia; 3Diagnostic Microbiology Department, King Faisal Hospital, Ministry of Health, Makkah 21955, Saudi Arabia; 4Medical Genetics Department, King Faisal Hospital, Ministry of Health, Makkah 21955, Saudi Arabia; 5Pharmaceutical Department, King Faisal Hospital, Ministry of Health, Makkah 21955, Saudi Arabia; 6Health Administration, King Faisal Hospital, Ministry of Health, Makkah 21955, Saudi Arabia

**Keywords:** antibiotics, coronavirus, intensive care unit, patients, resistance, Saudi Arabia

## Abstract

Antibiotic resistance is a global health and development threat, especially during the Severe Acute Respiratory Syndrome Coronavirus 2 (COVID-19) pandemic. Therefore, the current study was conducted to describe antibiotic usage and resistance among patients with COVID-19 in the intensive care unit (ICU) in Makkah, Saudi Arabia. In this cross-sectional study, only patients with positive COVID-19 status (42 patients) admitted to the ICU at the King Faisal Hospital were selected using a census sampling method. The susceptibility test of bacteria was carried out according to the standard protocol. The identified strains were tested in-vitro against several antibiotics drugs. Statistical analysis was performed using SPSS version 24. A total of 42 patients were included, with a mean age of 59.35 ± 18 years. Of them, 38.1% were males, and 61.9% were females. 35.7% have blood group O +. For age and blood groups, statistically significant associations were found between males and females, with *p*-values = 0.037 and 0.031, respectively. A large percentage (42.7%) of the obtained samples contained Klebsiella Pneumoniae; all bacteria were multidrug-resistance bacteria. Furthermore, 76.2% of bacteria were resistant to Ampicillin, 66.7% were resistant to Ciprofloxacin, 64.3% were resistant to Levofloxacin, 57.1% were resistant to Imipenem, and 57.1% were resistant to Moxifloxacin. On the contrary, among the 40 examined antibiotics, the effective antibiotics were Daptomycin, Linezolid, Mupirocin, Synercid, Teicoplanin, Vancomycin, and Nitrofurantoin. Our study demonstrates that antibiotic resistance is highly prevalent among ICU patients with COVID-19 at the King Faisal Hospital. Additionally, all bacteria were multidrug-resistance bacteria. Therefore, this high prevalence should be seriously discussed and urgently considered.

## 1. Introduction

A challenge to global health and development is antimicrobial resistance (AMR), which includes antibiotic resistance [1]. One of the top 10 worldwide public health hazards to humanity, according to the World Health Organization (WHO), is AMR [2]. AMR mortality is predicted to surpass that of cancer and cardiovascular disease combined by 2050 [3]. Particularly, antibiotic resistance is brought on by drugs that are exclusively effective against specific bacterial parts. Since the medicine is highly selective, a change in these molecules will prevent or negate the drug’s destructive activity, resulting in antibiotic resistance [4]. In addition to changing the enzyme that antibiotics target, bacteria are also capable of using enzymes to change the antibiotic itself and thereby neutralize it [5]. As a result of drug resistance, antibiotics and other antimicrobial medicines become ineffective, and infections become increasingly difficult or impossible to treat [6]. In addition, misuse and overuse of antibiotics are the main drivers in the development of drug-resistant pathogens [7]. For instance, Saudi Arabia is one country where the public’s use of over-the-counter antibiotics without a prescription plays a significant role in how they behave [8]. Findings from a study conducted in Riyadh, Saudi Arabia, showed that antibiotics could readily be obtained without a prescription in 78% of pharmacies [9]. Without efficient antibiotics, even simple surgeries and regular treatments could become high-risk procedures, extending disease duration and, eventually, increasing the chance of premature death [10]. Additionally, antibiotic resistance has a significant financial impact on the economy. The prolonged disease causes death and incapacity, longer hospital admissions, the need for more expensive medications, and financial difficulties for those affected [11].

A recent study found that antibiotic-resistant bacteria cause 5 million indirect deaths and 1.3 million direct deaths annually. The projections were generated in 2019, before the COVID-19 pandemic that caused Severe Acute Respiratory Syndrome (SARS) aggravated the issue [12]. Unfortunately, those who are the most vulnerable to COVID-19 are also the most vulnerable to drug-resistant infections [13]. Although the world is struggling to control the COVID-19 pandemic, the development of AMR outbreaks should be considered [14]. Bacterial co-infection during viral infections is a significant cause of morbidity and mortality. However, the clinical evidence suggests that bacterial co-infection rates for COVID-19 patients are very low, but antibiotic prescribing remains high [15]. It was noted that about 72% of COVID-19 patients were treated with antibiotics even when not clinically indicated, and this heavy use of empiric antibiotics led to a high rate of AMR [16]. Furthermore, AMR is regarded as a significant factor in predicting patient outcomes and overall resource utilization following infections in ICU [17]. Some studies have shown that antibiotics have been administered to the majority of hospitalized COVID-19 patients and 80–100% of COVID-19 patients in the ICU [18,19]. Recently, studies showed a significant increase in AMR resulting from the COVID-19 pandemic [20,21]. ICUs are facing an emergency and spreading antibiotic-resistant bacterial strains all over the world. Moreover, some resistant bacterial strains have few treatment options [22]. Therefore, the current study was conducted to describe the antibiotics usage and resistance among patients with COVID-19 in the intensive care unit (ICU) at the King Faisal Hospital in Makkah, Saudi Arabia in Makkah, Saudi Arabia.

## 2. Methods

### 2.1. Study Design, Setting, and Period

This prospective cross-sectional study was conducted between 1 November 2020 and 31 January 2021 at the King Faisal Hospital in Makkah, Saudi Arabia, which includes 300 beds, serving 40 k patients monthly, more than 500,000 patients yearly, and the isolated ICU includes 30 beds.

### 2.2. Data Collection Procedure

Patients who were hospitalized in the ICU provided samples. Abscess, ascites fluid, blood, pleural fluid, nasal swabs, sputum, urine culture, and wound samples were taken from various sites. All samples were collected by hospital nurses using accepted techniques. Additionally, the Ministry of Health guideline was followed for blood-taking procedures, which calls for washing the skin with 2% chlorhexidine and a 70% isopropyl alcohol applicator for 30 s while employing a back-and-forth scrubbing motion. Additionally, two collections of blood cultures from patients were taken in order to rule out contamination. A reference lab received the materials for molecular identification.

### 2.3. Sample Size and Sampling

As all eligible patients were located during the study period, the census sampling approach was used to choose the study participants (between 1 November 2020 and 31 January 2021). All patients admitted to the King Faisal Hospital’s ICU during the study period, both genders, non-intubated patients who were spontaneously breathing, patients who were using antibiotics (narrow or broad spectrum) during the study period, patients of all ages with bacterial infections, and patients with a positive COVID-19 test were included in the study. Outpatients, patients with a negative COVID-19 test, and patients with other kinds of bacteria were all disqualified from the study. Each patient also provided one sample for a culture, and every single one of these cultures revealed the presence of germs.

### 2.4. Isolation and Identification of Pathogens

The ICU samples were handled by the King Faisal Hospital Microbiology Laboratory in accordance with the established protocols for bacterial isolation and identification. Blood culture bottles were incubated with Biomeieux BACT/ALERT. Once the machine indicated growth, the samples were cultured in blood, MacConkey, and chocolate agar. Grown-on blood, MacConkey, or CLED agar were urine sample samples. They spent the next five to seven days incubating. Preliminary identification of certain isolates was made using colony morphology, Gram stain, and typical rapid biochemical assays such as catalase, indole, and oxidase tests. According to the procedures of the hospital where the strain originated, gram-negative bacteria were found using Pos Breakpoint Combo Panel Type 50 in MicroScan (Beckman Coulter Inc., Brea, CA, USA) and gram-positive strains using Pos Breakpoint Combo Panel Type 28 in MicroScan (Beckman Coulter Inc., CA, USA).

### 2.5. Antibiotics Susceptibility Testing

The susceptibility test was conducted in accordance with the Clinical and Laboratory Standards Institute’s recommendations (CLSI). By utilizing the MicroScan automated microbiology technology, identified bacteria were evaluated in vitro against many types of antibiotic medications (Pos Breakpoint Combo 50 Panel). The following antibiotic agents were examined: Amikacin, Amoxicillin/Clavulanate, Sulbactam, Ampicillin, Aztreonam, Cefazolin, Cefepime, Cefotaxime, Cefoxitin, Ceftazidime, Ciprofloxacin, Cefuroxime, Colistin, Ertapenem, Gentamicin, Imipenem, Levofloxacin, Meropenem, Moxifloxacin, Piperacillin/Tazobactam, Tigecycline, Tobramycin, Trimethoprim/Sulfamethoxazole, Azithromycin, Clindamycin, Daptomycin, Erythromycin, Fosfomycin, Fusidic Acid, Linezolid, Mupirocin, Oxacillin, Penicillin, Rifampin, Synercid, Teicoplanin, Tetracycline, Vancomycin, Nitrofurantoin, and Norfloxacin. Quality control and maintenance were achieved according to the manufacturer’s guidelines.

### 2.6. Ethical Considerations

The College of Medicine at Umm Al Qura University’s Research and Ethical Committee gave its approval to the study protocol (HAPO-02-K-012-2021-08-713). Additionally, consent was obtained from the King Faisal Hospital. Conscious patients and the representatives of unconscious patients who agreed to take part in the study were required to complete a written informed consent form.

### 2.7. Statistical Analysis

Data analysis was done using the Statistical Package for Social Science (SPSS) version 24 (IBM Corp, Armonk, NY, USA). For continuous variables, data are expressed as means and standard deviation; for categorical variables, they are expressed as a percentage. The independent sample t-test was used to examine any variations in means. The prevalence of several categorical variables was compared using the chi-square test. A *p*-value of 0.05 or less was regarded as statistically significant.

## 3. Results

In the current study, out of 298 patients who were admitted to the ICU during the study period, only 42 patients with positive COVID-19 tests were included in the final analysis. Of them, 16 (38.1%) were males, and 26 (61.9%) were females (Figure 1). The mean age (years) for the study participants was 59.35 ± 18 (66.62 ± 15 for males and 54.88 ± 20 for females). The results revealed that 15 (35.7%), 9 (21.45), 9 (21.4%), 3 (7.1%), 2 (4.8%), 2 (4.8%), and 2 (4.8%) of them have blood group O+, O−, A+, A−, AB+, B+, and B− respectively. A large percentage of 12 (46.2%) female patients have blood group O +, while only 3 (18.8%) males have blood group O+. Concerning medical diagnosis of the patients, the results showed that 2 (4.8%) of the patients have an acute myocardial infarction, 4 (9.5%) have acute pain, 1 (2.4%) have chronic kidney disease, 1 (2.4%) have dyspnea, 4 (9.5%) have a heart attack, 4 (9.5%) have pneumonia, 3 (7.1%) have a stroke, 13 (30.9%) have sepsis, 1 (2.4%) have weakness, 7 (16.7%) have an unknown fever, and 2 (4.8%) have a viral infection.

In addition, the results demonstrated that 23 (54.8%) of the patients were discharged from the hospital after treatments, and 19 (45.25%) passed away. Furthermore, for age and blood groups, statistically significant associations were found between males and females, with *p*-values = 0.037 and 0.031, respectively (Table 1).

A total of 42 samples were collected from the patients. Of them, 12 (28.6%) were from blood, 10 (23.8%) from sputum, 9 (21.4%) from a urine culture, 6 (14.2%) from wounds, 2 (4.8%) from nasal swabs, 1 (2.4%) from an abscess, 1 (2.4%) from ascites fluid, and 1 (2.4%) from pleural fluid. Concerning the types of bacteria, the findings show that 18 (42.7%) of the samples contain *Klebsiella Pneumoniae*, 4 (9.5%) *Methicillin Resistant Staphylococcus Aureus*, 3 (7.1%) *Acinetobacter Baumannii Complex/Hemolyticus*, 3 (7.1%) *Pseudomonas Aeruginosa*, 2 (4.8%) *Escherichia Coli*, 2 (4.8%) *Escherichia Coli ESBL*, 2 (4.8%) *Proteus Mirabilis*, 2 (4.8%) *Staphylococcus Aureus*, 2 (4.8%) *Staphylococcus Epidermidis*, 2 (4.8%) *Staphylococcus Hominis subspecies Hominis*, 1 (2.4%) *Staphylococcus Hemolyticus*, and 1 (2.4%) contains *Streptococcus agalactiae*. No statistically significant association was found between both genders (Table 2).

Table 3 shows the types of bacteria by the sources of the samples. The results revealed that 18 (42.7%) of the samples contained *Klebsiella Pneumoniae* and were obtained from sputum n = 6 (60%), urine culture n = 4 (44.5%), blood n = 3 (25.1%), wound n = 2 (33.3%), body fluid n = 1 (100%), ascites fluid n = 1 (100%), and abscess culture n = 1 (100%). 1 (2.4%) of the samples contained *Staphylococcus Hemolyticus*, and 1 (2.4%) contained *Streptococcus agalactiae* were obtained from blood and urine cultures. In addition, 12 (28.6%) of the blood samples contain bacteria, 10 (23.8%) of the sputum samples contain bacteria, 9 (21.4%) of the urine cultures contain bacteria, and 6 (14.2%) of the wound cultures contain bacteria.

Moreover, Table 4 shows the resistance pattern to the most commonly used antibiotics by the types of bacteria among the study participants. The findings show that all bacteria in the current study, such as *Acinetobacter Baumannii Complex/Hemolyticus*, *Escherichia Coli*, *Escherichia Coli ESBL*, *Klebsiella Pneumoniae*, *Methicillin Resistant Staphylococcus Aureus*, *Proteus Mirabilis*, *Pseudomonas Aeruginosa*, *Staphylococcus Aureus*, *Staphylococcus Epidermidis*, *Staphylococcus Hemolyticus*, *Staphylococcus Hominis* subspecies *Hominis*, and *Streptococcus agalactiae* were multidrug-resistance bacteria. In addition, 32 (76.2%) of bacteria were resistant to Ampicillin, 28 (66.7%) were resistant to Ciprofloxacin, 27 (64.3%) were resistant to Levofloxacin, 24 (57.1%) were resistant to Imipenem, and 24 (57.1%) were resistant to Moxifloxacin.

On the contrary, among the 40 examined antibiotics (Amikacin, Amoxicillin/Clavulanate, Sulbactam, Ampicillin, Aztreonam, Cefazolin, Cefepime, Cefotaxime, Cefoxitin, Ceftazidime, Ciprofloxacin, Cefuroxime, Colistin, Ertapenem, Gentamicin, Imipenem, Levofloxacin, Meropenem, Moxifloxacin, Piperacillin/Tazobactam, Tigecycline, Tobramycin, Trimethoprim/Sulfamethoxazole, Azithromycin, Clindamycin, Daptomycin, Erythromycin, Fosfomycin, Fusidic Acid, Linezolid, Mupirocin, Oxacillin, Penicillin, Rifampin, Synercid, Teicoplanin, Tetracycline, Vancomycin, Nitrofurantoin, and Norfloxacin), the effective antibiotics against the included bacteria were Daptomycin, Linezolid, Mupirocin, Synercid, Teicoplanin, Vancomycin, and Nitrofurantoin.

## 4. Discussion

In this cross-sectional study, the susceptibility test of bacteria was carried out according to the standard protocol, and the identified strains were tested in-vitro against several antibiotics drugs among 42 patients with positive COVID-19 who were selected using a census sampling method and admitted to the ICU at the King Faisal Hospital. The current study included all patients with bacterial infections, all genders, non-intubated spontaneously breathing patients utilizing antibiotics (narrow or broad spectrum) during the study period, all ages, and positive COVID-19 test results who were admitted to the ICU during the study period.

Our study was not intended to investigate whether the interruption in hospital operations during the pandemic crisis or COVID-19, in general, is associated with the probability of MDRB acquisition. MDRB spreads by cross-transmission or environmental factors in the hospital context, with pressure from antimicrobial therapy selection favoring certain individuals [23]. Due to the frequent and complicated caring, which makes it easier for the contamination of health care personnel’s hands and, as a result, the spread of MDRB [23], cross-transmission occurs in the ICU environment in between 23% and 53% of patient encounters [24,25]. Key elements in preventing the spread of MDRB are the implementation of infection prevention and control measures and the supervision of their observance [26].

Since many earlier pandemics occurred before the period of antibiotic resistance, no information was available prior to the SARS-CoV-2 pandemic, including information on the spread of MDRB during the H1N1 pandemic [27]. Concerns concerning the spread of MDRB during the COVID-19 pandemic have been expressed by experts [28,29,30], and a report suggests a rise in bloodstream infection [31].

Our study showed that a large percentage (42.7%) of the obtained samples contained *Klebsiella Pneumoniae*, and all bacteria were multidrug-resistance bacteria highlighting the urgent need for the development of newer and more robust antimicrobial agents [32,33]. Additionally, 76.2% of bacteria were resistant to Ampicillin, 66.7% to Ciprofloxacin, 64.3% to Levofloxacin, 57.1% to Imipenem, and 57.1% to Moxifloxacin. The large rate of MDRB acquisition in COVID-19 patients is alarming, and this is in line with our concerns. Furthermore, it is important to remember that the following two elements ought to have lowered the MDRB acquisition rate: Patients with COVID-19 were admitted to ICUs that had been totally cleaned out and decontaminated, in contrast to normal ICU admission, which occurs in units where MDRB carriers are already present. A further effective defense against MDRB cross-transmission should have been the physical separation of COVID-19 patients.

The high rate of MDRB acquisition may be explained by a number of pandemic-related factors, including a lack of PPE [34], the ICU staff being overworked, the overcrowding of the ICU, and the reinforcement of less experienced staff, which reduced adherence to infection prevention and control measures [28,29,30,35]. The prevalence of MDRB was assessed in 2020 compared to the years 2017–2019, according to Aurilio C. et al. (2021); the prevalence of overall MDRB infection was 45.2% in 2017, 44.2% in 2018, 41.4% in 2019, 19.2% in 2020 in non–COVID–19 wards, and 29.3% in COVID–19 wards [36]. Although the purpose of our study was not to determine how each of these many systems functioned, we can make the following hypothesis to explain our results. First, there was a shortage of gowns, so we had to use the same gown on multiple patients. It is probable that gloves were not systematically removed at that time due to the difficulty of taking off PPE. This behavior was not observable in our investigation. A new four-bed ICU with two patients per room had to be created as a result of our study having to modify our standard single-room policy due to the spike in patients. A firm conclusion cannot be formed on this topic due to the small number of patients who were worried; however, staying in our four-bed ICU was not linked to MDRB acquisition. Third, we were unable to continue our audit of catheter dressing and hand hygiene practices as prevention and control measures.

A well-known factor linked to the acquisition of MDRBs is an increase in the use of antimicrobials, in addition to infection control measures [37]. Patients with COVID-19 were first thought to have a significant risk of bacterial co-infection and secondary nosocomial infections, similar to patients with other viral illnesses [38]. Additionally, even in cases when there is no bacterial infection, the early COVID-19 symptoms may encourage the start of antibiotic therapy [28]. Despite the possibility of confounding factors, our investigation demonstrated that, consistent with early findings [39], excessive antibiotic use was significantly related to a higher probability of MDRB acquisition in COVID-19 individuals. It is yet unclear if using broad-spectrum classes alone carries this risk or whether using any antibiotics at all.

Finally, throughout the initial wave, we avoided using immunosuppressive medications. The administration of dexamethasone or the other immunosuppressive medications under study may theoretically raise the likelihood of acquiring MDRB even further.

Many investigations were carried out in ICU settings, which quantitatively showed higher rates of AMR than in non-ICU settings. Despite being influenced by two cohort studies, it is not unusual for AMR to be found in substantially higher concentrations in ICU settings. Prior to COVID-19, patients admitted to ICU settings were more likely to contract infections; studies have shown that 20–50% of ICU hospitalizations were for nosocomial infections [40,41,42]. Higher co-infection rates, particularly those of a resistant nature, are not surprising given the COVID-19 pandemic, where patients are a priori given a combination of antimicrobial and immunosuppressive agents. This is especially true in patients who have been mechanically ventilated for extended periods of time. Furthermore, geographical differences that increase the risk of contracting AMR infections, particularly in low- and middle-income countries, high livestock and food product movement, poor clean water and sanitation facilities, as well as a lack of routine surveillance in these areas all contribute to the overall increase in AMR [43].

The misuse of antibiotics in hospitals may also be a contributing factor to antibiotic resistance; it is estimated that 25 to 50 percent of the antimicrobials prescribed in hospitals are unnecessary or inappropriate, directly affecting AMR [41]. Furthermore, the capacity of the pharmaceutical industry to release new antimicrobials onto the market is inferior to the capacity of microorganisms to develop resistance to previously susceptible drugs [42]. Since the start of the COVID-19 pandemic, scientists have cautioned against the risks of antibiotic abuse, despite clinical evidence with prior viral epidemics suggesting concerns of bacterial co-infection [43,44]. Significant antimicrobial use is anticipated at the hospital’s intensive care unit (ICU) due to the seriousness of the diseases treated and the number of interventions given to patients. Antimicrobial monitoring is therefore essential in the event of a pandemic to spot alarming indicators of abuse or overuse.

In some settings, surveillance programs, thorough testing with standardized protocols and reporting, as well as multimodal strategies emphasizing the strict use of antibiotics in conjunction with infection, prevention, and control practices, could improve antimicrobial stewardship, ultimately lowering mortality and morbidity, particularly in COVID-19 patients. Recent research suggests that these multimodal approaches can be very successful in preventing the spread of resistant microbes [44,45].

It has been determined which clinical and sociodemographic factors, such as health care settings, socioeconomic level, prior antibiotic use, and length of hospital stay, increase a patient’s chance of acquiring such co-infections. Particularly in the case of SARS-CoV-2, identifying individuals who are at higher risk of getting MDR or XDR infections beforehand may improve overall prognosis and outcomes. Additionally, some earlier studies [46,47,48,49] advocate the use of natural substances as fresh methods to combat AMR. We could better understand AMR during COVID-19 and in the future with prospective studies that combine well-designed microbiological investigations [50].

Our study showed that 35.7% have blood group O +. Similar findings were reported by Shesha and her colleagues in Saudi Arabia among non-ICU-admitted COVID-19 patients [51].

The main strength of our study was it is the first study that shows the antibiotics usage and resistance among patients with COVID-19 in the ICU at the King Faisal Hospital in Makkah, Saudi Arabia. The main limitation of this study is its cross-sectional design, which restricts the applicability of our findings. Additionally, there was no follow-up for the study participants because the study was cross-sectional. Additionally, all of the study isolates were obtained from inpatients, and it was impossible to distinguish between the precise amount of nosocomial and community-acquired bacteria. It is advised that these findings be confirmed by other multicenter trials with high sample sizes conducted over extended time periods.

## 5. Conclusions

In conclusion, our study demonstrates that antibiotic resistance is highly prevalent among ICU patients with COVID-19 at the King Faisal Hospital. Additionally, all of the obtained bacteria were multidrug-resistance bacteria. Furthermore, among the 40 examined antibiotics, the effective antibiotics were Daptomycin, Linezolid, Mupirocin, Synercid, Teicoplanin, Vancomycin, and Nitrofurantoin. Therefore, this high prevalence should be seriously discussed and urgently considered. Additionally, more creativity and funding are needed for operational research as well as for the development of novel antimicrobial drugs, vaccines, and diagnostic tools, particularly those that target dangerous gram-negative bacteria.

## Figures and Tables

**Figure 1 vaccines-10-02148-f001:**
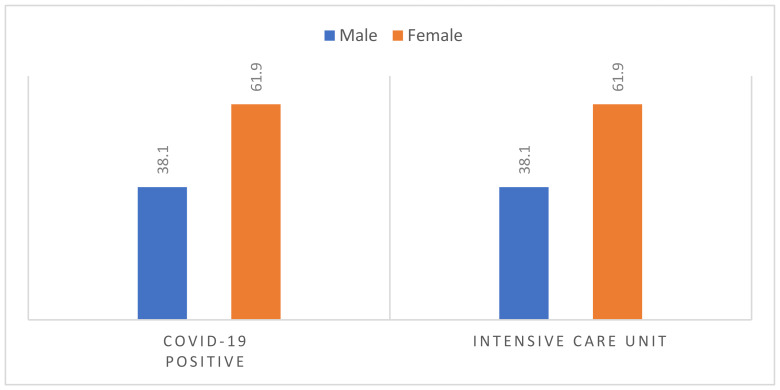
Distribution of the study participants according to the results of the COVID-19 test and departments by gender.

**Table 1 vaccines-10-02148-t001:** Characteristics and medical history variables of the study participants by gender.

Variables	Total: *n* = 42 (%)	Male: *n* = 16 (%)	Female: *n* = 26 (%)	*p* Value
Age (years)				
Mean ± SD	59.35 ± 18	66.62 ± 15	54.88 ± 20	0.037
Blood group				
A−	2 (4.8)	2 (100)	0.0 (0.0)	0.031
A+	9 (21.4)	2 (22.2)	7 (77.8)	
AB+	2 (4.8)	2 (100)	0.0 (0.0)	
B−	2 (4.8)	2 (100)	0.0 (0.0)	
B+	3 (7.1)	2 (66.7)	1 (33.3)	
O−	9 (21.4)	3 (33.3)	6 (66.7)	
O+	15 (35.7)	3 (20.0)	12 (80.0)	
Diagnosis				
Acute myocardial infarction	2 (4.8)	2 (100)	0.0 (0.0)	0.085
Acute pain	4 (9.5)	0.0 (0.0)	4 (100)	
Chronic kidney disease	1 (2.4)	0.0 (0.0)	1 (100)	
Dyspnea	1 (2.4)	0.0 (0.0)	1 (100)	
Heart attack	4 (9.5)	3 (75.0)	1 (25.0)	
Pneumonia	4 (9.5)	1 (25.0)	3 (75.0)	
Stroke	3 (7.1)	3 (100)	0.0 (0.0)	
Sepsis	13 (30.9)	5 (38.5)	8 (61.5)	
Weakness	1 (2.4)	0.0 (0.0)	1 (100)	
Unknown fever	7 (16.7)	2 (28.6)	5 (71.4)	
Viral infection	2 (4.8)	0.0 (0.0)	2 (100)	
Outcome				
Discharge	23 (54.8)	6 (26.1)	17 (73.9)	0.074
Passed away	19 (45.2)	10 (52.6)	9 (47.4)	

Data are expressed as means ± SD for continuous variables and as a percentage for categorical variables. The differences between means were tested by using the independent sample *t*-test. The chi-square test was used to examine differences in the prevalence of different categorical variables. A *p*-value less than 0.05 was considered statistically significant. SD, stander deviation.

**Table 2 vaccines-10-02148-t002:** The source of samples and types of bacteria among the study participants by gender.

Variables	Total: *n* = 42 (%)	Male: *n* = 16 (%)	Female: *n* = 26 (%)	*p* Value
Source of samples				
Abscess	1 (2.4)	0.0 (0.0)	1 (100)	0.581
Ascites fluid	1 (2.4)	0.0 (0.0)	1 (100)	
Blood	12 (28.6)	7 (58.3)	5 (41.7)	
Pleural fluid	1 (2.4)	0.0 (0.0)	1 (100)	
Nasal swabs	2 (4.8)	1 (50.0)	1 (50.0)	
Sputum	10 (23.8)	3 (30.0)	7 (70.0)	
Urine culture	9 (21.4)	4 (44.4)	5 (55.6)	
Wound	6 (14.2)	1 (16.7)	5 (83.3)	
Types of bacteria				
*Acinetobacter Baumannii Complex/Hemolyticus*	3 (7.1)	2 (66.7)	1 (33.3)	0.277
*Escherichia coli*	2 (4.8)	0.0 (0.0)	2 (100)	
*Escherichia coli ESBL*	2 (4.8)	1 (50.0)	1 (50.0)	
*Klebsiella pneumoniae*	18 (42.7)	5 (27.8)	13 (72.2)	
*Methicillin-Resistant Staphylococcus Aureus*	4 (9.5)	2 (50.0)	2 (50.0)	
*Proteus Mirabilis*	2 (4.8)	0.0 (0.0)	2 (100)	
*Pseudomonas Aeruginosa*	3 (7.1)	0.0 (0.0)	3 (100)	
*Staphylococcus Aureus*	2 (4.8)	1 (50.0)	1 (50.0)	
*Staphylococcus Epidermidis*	2 (4.8)	1 (50.0)	1 (50.0)	
*Staphylococcus Hemolyticus*	1 (2.4)	1 (100)	0.0 (0.0)	
*Staphylococcus Hominis* subspecies *Hominis*	2 (4.8)	2 (100)	0.0 (0.0)	
*Streptococcus agalactiae*	1 (2.4)	1 (100)	0.0 (0.0)	

In the case of categorical variables, data are presented as a percentage. The prevalence of several categorical variables was compared using the chi-square test. A *p*-value of 0.05 or less was regarded as statistically significant.

**Table 3 vaccines-10-02148-t003:** Types of bacteria by the sources of the samples.

Types of Bacteria	Abscess	Ascites Fluid	Blood	Fluid	Nasal Swap	Sputum	Urine Culture	Wound	Total: *n* (%)
*Acinetobacter Baumannii Complex/Hemolyticus*	-	-	1 (8.3)	-	-	1 (10)	1 (11.1)	-	3 (7.1)
*Escherichia coli*	-	-	-	-	-	-	2 (22.2)	-	2 (4.8)
*Escherichia coli ESBL*	-	-	1 (8.3)	-	-	1 (10)	-	-	2 (4.8)
*Klebsiella Pneumoniae*	1 (100)	1 (100)	3 (25.1)	1 (100)	-	6 (60)	4 (44.5)	2 (33.3)	18 (42.7)
*Methicillin-Resistant Staphylococcus Aureus*	-	-	-	-	2 (100)	-	-	2 (33.3)	4 (9.5)
*Proteus Mirabilis*	-	-	1 (8.3)	-	-	-	1 (11.1)	-	2 (4.8)
*Pseudomonas Aeruginosa*	-	-	-	-	-	2 (20)	-	1 (16.7)	3 (7.1)
*Staphylococcus Aureus*	-	-	1 (8.3)	-	-	-	-	1 (16.7)	2 (4.8)
*Staphylococcus Epidermidis*	-	-	2 (16.7)	-	-	-	-	-	2 (4.8)
*Staphylococcus Hemolyticus*	-	-	1 (8.3)	-	-	-	-	-	1 (2.4)
*Staphylococcus Hominis* subspecies *Hominis*	-	-	2 (16.7)	-	-	-	-	-	2 (4.8)
*Streptococcus agalactiae*	-	-	-	-	-	-	1 (11.1)	-	1 (2.4)
Total:	1 (2.4)	1 (2.4)	12 (28.6)	1 (2.4)	2 (4.8)	10 (23.8)	9 (21.4)	6 (14.2)	42 (100)

Data are expressed as a percentage of categorical variables.

**Table 4 vaccines-10-02148-t004:** Resistance to the most commonly used antibiotics by the types of bacteria among the study participants.

Types of Bacteria	Amikacin	Amoxicillin/Clavulanate	Sulbactam	Ampicillin	Aztreonam	Cefazolin	Cefepime	Cefotaxime	Cefoxitin	Ceftazidime	Ciprofloxacin	Cefuroxime	Colistin	Ertapenem	Gentamicin	Imipenem	Levofloxacin	Meropenem	Moxifloxacin	Piperacillin/Tazobactam	Tigecycline	Tobramycin	Trimethoprim/Sulfamethoxazole	Azithromycin	Clindamycin	Daptomycin	Erythromycin	Fosfomycin	Fusidic Acid	Linezolid	Mupirocin	Oxacillin	Penicillin	Rifampin	Synercid	Teicoplanin	Tetracycline	Vancomycin	Nitrofurantoin	Norfloxacin
*Acinetobacter Baumannii Complex/Hemolyticus: n* = 3	3	-	3	3	-	-	3	3	-	3	3	-	-	-	3	3	3	3	-	-	-	3	3	-	-	-	-	-	-	-	-	-	-	-	-	-	-	-	-	-
*Escherichia Coli: n* = 2	-	-	-	-	-	-	-	-	-	-	-	-	-	-	-	-	-	-	-	-	-	-	-	-	-	-	-	-	-	-	-	-	-	-	-	-	-	-	-	-
*Escherichia Coli ESBL: n* = 2	-	-	-	2	-	2	2	-	-	2	1	2	-	-	-	-	1	-	2	-	-	-	-	-	-	-	-	-	-	-	-	-	-	-	-	-	-	-	-	-
*Klebsiella Pneumoniae: n* = 18	14	13	14	17	13	1	14	14	13	13	15	14	-	13	14	11	14	13	15	13	2	14	15	-	-	-	-	-	-	-	-	-	-	-	-	-	-	-	-	-
*Methicillin Resistant Staphylococcus Aureus: n* = 4	-	4	-	4	-	-	-	-	-	-	3	-	-	-	-	4	3	-	3	-	-	-	-	-	-	-	-	-	-	-	-	4	4	-	-	-	-	-	-	-
*Proteus Mirabilis: n* = 2	-	-	-	-	-	-	-	-	-	-	-	-	-	-	-	-	-	-	-	-	-	-	-	-	-	-	-	-	-	-	-	-	-	-	-	-	-	-	1	2
*Pseudomonas Aeruginosa: n* = 3	-	-	-	-	1	-	-	-	-	-	-	-	1	-	-	-	-	-	-	-	-	-	-	-	-	-	-	-	-	-	-	-	-	-	-	-	-	-	-	-
*Staphylococcus Aureus: n* = 2	2	-	-	2	-	-	-	-	-	-	2	-	-	-	-	2	2	-	-	-	-	-	-	2	1	-	2	-	1	-	-	2	2	-	-	-	-	-	-	-
*Staphylococcus Epidermidis: n* = 2	2	2	-	2	-	-	-	-	-	-	2	-	-	-	2	2	2	-	2	-	-	-	-	-	-	-	2	-	-	-	-	2	2	2	-	-	-	-	-	-
*Staphylococcus Hemolyticus: n* = 1	1	-	-	1	-	-	-	-	-	-	1	-	-	-	1	1	1	-	1	-	-	-	-	1	1	-	1	1	1	-	-	1	1	-	-	-	1	-	-	-
*Staphylococcus Hominis subspecies Hominis: n* = 2	-	1	-	1	-	-	-	-	-	-	1	-	-	-	1	1	1	-	1	-	-	-	-	1	1	-	1	-	1	-	-	1	1	-	-	-	-	-	-	-
*Streptococcus Agalactiae: n* = 1	-	-	-	-	-	-	-	-	-	-	-	-	-	-	-	-	-	-	-	-	-	-	-	-	-	-	-	-	-	-	-	-	-	-	-	-	1	-	-	-
Total: 42 (100%)	22 (52.4%)	20 (47.6%)	17 (40.5%)	32 (76.2%)	14 (33.3%)	3 (7.14%)	19 (45.2%)	17 (40.5%)	13 (30.9%)	18 (42.8%)	28 (66.7%)	16 (38.1%)	1 (2.38%)	13 (30.9%)	21 (50.0%)	24 (57.1%)	27 (64.3%)	16 (38.1%)	24 (57.1%)	13 (30.9%)	2 (4.76%)	17 (40.5%)	18 (42.8%)	4 (9.52%)	3 (7.14%)	0.0 (0.0%)	6 (14.28%)	1 (2.38%)	3 (7.14%)	0.0 (0.0%)	0.0 (0.0%)	10 (23.8%)	10 (23.8%)	2 (4.76%)	0.0 (0.0%)	0.0 (0.0%)	2 (4.76%)	0.0 (0.0%)	1 (2.38%)	2 (4.76%)

## Data Availability

The data presented in this study are available on request from the corresponding author. The data are not publicly available due to privacy.

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
