# Peer review of "Antibiotics Usage and Resistance among Patients with Severe Acute Respiratory Syndrome Coronavirus 2 in the Intensive Care Unit in Makkah, Saudi Arabia"

_vaccines, 2022, doi:10.3390/vaccines10122148_

Round 1

Reviewer 1 Report

The manuscript entitled "Antibiotics Usage and Resistance among Patients with Severe Acute Respiratory Syndrome Coronavirus 2 in the Intensive Care Unit in Makkah, Saudi Arabia" described the antibiotic usage and resistance among patients with COVID-19 in the intensive care unit (ICU) in Makkah, Saudi Arabia. This reviewer believed that it would be great to introduce some new approaches for dealing with antimicrobial resistance. For example, use of natural compounds is increasingly considered as a new direction to deal with antimicrobial resistance. Some of previous excellent studies demonstrated that these compounds could be used for antibiotic resistant organisms, e.g., PMID: 34352473, PMID: 34521050, PMID: 33310352, PMID: 33529791,PMID: 32818706, PMID: 33017769, PMID: 33302226, PMID: 32416447, PMID: 32798979 and should be cited in this study. Introduction of new approaches for antimicrobial resistance would significantly promote readership of this review. It would also be very important to include the presence and distribution of antibiotic resistance genes in animal gut microbiota (PMID: 34742104 ) and in humans (PMID: 34592562). 

I have the following remarks to improve the manuscript:

1. The conditions of the patients should be described in detail.

2. How the pathogens were identified and verified should be described in detail.

3. The authors should also discuss why bacteria are highly resistant to multiple antibiotics in patients with COVID in ICU.

4. What was the inclusion criteria and exclusion criteria of the subjects in this study?

5. Some of the tables could be converted into figures.

Author Response

Comments and Suggestions for Authors

The manuscript entitled "Antibiotics Usage and Resistance among Patients with Severe Acute Respiratory Syndrome Coronavirus 2 in the Intensive Care Unit in Makkah, Saudi Arabia" described the antibiotic usage and resistance among patients with COVID-19 in the intensive care unit (ICU) in Makkah, Saudi Arabia. This reviewer believed that it would be great to introduce some new approaches for dealing with antimicrobial resistance. For example, use of natural compounds is increasingly considered as a new direction to deal with antimicrobial resistance. Some of previous excellent studies demonstrated that these compounds could be used for antibiotic resistant organisms, e.g., PMID: 34352473, PMID: 34521050, PMID: 33310352, PMID: 33529791,PMID: 32818706, PMID: 33017769, PMID: 33302226, PMID: 32416447, PMID: 32798979 and should be cited in this study. Introduction of new approaches for antimicrobial resistance would significantly promote readership of this review. It would also be very important to include the presence and distribution of antibiotic resistance genes in animal gut microbiota (PMID: 34742104 ) and in humans (PMID: 34592562). 

We used the suggester studies for antibiotic resistant organisms.

I have the following remarks to improve the manuscript:

  1. The conditions of the patients should be described in detail.

Thank you for this important point. The conditions of the patients were described in detail.

  1. How the pathogens were identified and verified should be described in detail.

Biomeieux BACT/ALERT was used to incubate blood culture bottles. The samples were grown in blood, MacConkey, and chocolate agar once the machine signaled growth. Samples of urine were grown on blood, MacConkey, or CLED agar. After that, they have spent 5 to 7 days incubating. Based on colony morphology, Gram stain, and common fast biochemical assays such catalase, indole, and oxidase tests, preliminary identification of certain isolates was carried out.  Gram-negative bacteria were detected by Pos Breakpoint Combo Panel Type 50 in MicroScan (Beckman Coulter Inc., CA, United States), and gram-positive strains by Pos Breakpoint Combo Panel Type 28 in MicroScan, in accordance with the protocols of the hospital where the strain originated (Beckman Coulter Inc., CA, United States).

  1. The authors should also discuss why bacteria are highly resistant to multiple antibiotics in patients with COVID in ICU.

We have further discussed why bacteria are highly resistant to multiple antibiotics in patients with COVID in ICU.

  1. What was the inclusion criteria and exclusion criteria of the subjects in this study?

The study participants were selected using the census sampling method as all eligible patients were found during the study period (between November 1, 2020, and January 31, 2021). The inclusion criteria include all patients who were admitted to the ICU at the King Faisal Hospital during the study period, both genders, all ages, with bacterial infection and who have a positive COVID-19 test were included in the study. The exclusion criteria include outpatients, patients with a negative COVID-19 test and patients with other types of microorganisms were excluded from the study. In the currents study, out of 298 patients who were admitted to the ICU during the study period, only 42 patients with positive COVID-19 test were included in the final analysis. Additionally, one culture sample was obtained from each patient according to his/her medical condition.

  1. Some of the tables could be converted into figures.

We have added figure no. 1.

Reviewer 2 Report

Review: 

 Antibiotics Usage and Resistance among Patients with Severe Acute Respiratory Syndrome Coronavirus 2 in the Intensive Care Unit in Makkah, Saudi Arabia

Introduction: 

Therefore, the current study was conducted to describe the antibiotics usage and resistance among patients with COVID-19 in the intensive care unit (ICU) at the King Faisal Hospital in Makkah, Saudi Arabia in Makkah, Saudi Arabia.

I would like to hear more of 20,21 ref in the background- was it done in the ICU? If yes – what is new about your effort, or please specify the novelty of your research.

If an English native speaker was not reviewing the text – please try to review phrases again: 

Although the world is suffering to control the COVID-19 pandemic, the development of AMR outbreaks development should be considered.

2. METHODS 

2.1. Study design, setting, and period 

This cross-sectional study was conducted at the King Faisal Hospital in Makkah, Saudi Arabia, between November 1, 2020, and January 31, 2021.

Prospective? Retrospective? 

Only three months? How much perspective is there? 

Hospital – how many beds?  In the hospital – how big is the population it serves, and how many icus? Who are the patients? Ards patients? What were the criteria for them as ards patients?  

Data collection: 

No need to remind the name and place of the hospital, city, or country so many times. 

Before describing the data collection, please specify – inclusion exclusions.

Primary and secondary outcomes 

Samples from: body fluids? ? please specify- ascites is body fluid. Pleural effusions? Others? 

Nasal swaps? Swabs 

Under sample size – “298 patients admitted to the ICU were eligible to be included in the study.

This should be part of the results, not the methods. 

Only patients with positive COVID-19 (42 patients) were included in this study.--> what kind of patients? Intubated none? How sick? Under ab or not? Please describe the population – it is a highly heterogeneous one. 

One sample was obtained from each patient according to his/her medical condition.--> please specify what it means according to a medical condition – definitions should be well described so that others can know who the studied population was and how to repeat your study somewhere else… 

2.7. Statistical analysis 

The Statistical Package for Social Science (SPSS) version 24 (IBM Corp, Armonk, NY, USA) was used for data analysis, including descriptive statistics of frequencies and percentages.-

Not enough- what were the methods for analysis? Comparisons? Statistical tests? Etc.. 

3. RESULTS 

A total of 42 patients with COVID-19 who were admitted to the ICU at the King Faisal Hospital in Makkah, Saudi Arabia, were included in the current study.

Please explain what has happened from 298 eligible patients to only 42. 

Were these 42 positive for cultures? Unclear 

One sample from each patient? All 42 positive? 

Very small sample size – very hard to draw conclusions 

The text under table 1 : 

Data are expressed as means ± SD for continuous variables and as a percentage for categorical variables. The differences between means were tested by using the independent sample t-test. The chi-square test was used to examine differences in the prevalence of different categorical variables. P value less than 0.05 was considered statistically significant. SD, stander deviation.

Should go to statistics in methods 

Discussion: 

Despite the possibility of confounding factors, our investigation demonstrated that consistent with early findings [39], excessive antibiotic use was significantly related to a higher probability of MDRB acquisition in COVID-19 individuals. It is yet unclear if using broad-spectrum classes alone carries this risk or whether using any antibiotics at all.”  There was no presentation regarding how and what antibiotics were provided to these patients before – so this conclusion is problematic 

5. Conclusion 

In conclusion, our study demonstrates that antibiotic resistance is highly prevalent among ICU patients with COVID-19. Additionally, all of the obtained bacteria were multidrug-resistance bacteria.--> this is true in one specific hospital, not worldwide. 

Major: 

The sample size is a major factor here. Small numbers were collected in the very short time period in only one medical 

center 

the power of this study is very low as well as its generalizability 

not sure what conclusions can be drawn from it 

It is unclear what the ratio of positive cultures was – this 42 out of how many cultures? 

the study population is not well defined 

it is unclear if from 42 patients were taken only 42 cultures – one patient, one culture? If only patients that had growth of bacteria were included? How come we do not have more than one growth per patient? Other positive cultures were excluded. What was the median time in the ICU when these cultures were taken? What were the reasons for taking these cultures?

How many of them were on antibiotics? For how long? And how many? 

Were there any outbreaks of more in the ICU? 

The authors argue that they have a high rate of MDR infections – but are not comparing to any other data – what were the rates of MDR infection for covid patients in different places around the world with different antibiotic abuse rates?

All these limitations -were not discussed. 

Author Response

Comments and Suggestions for Authors

Review:

Antibiotics Usage and Resistance among Patients with Severe Acute Respiratory Syndrome Coronavirus 2 in the Intensive Care Unit in Makkah, Saudi Arabia

Introduction:

Therefore, the current study was conducted to describe the antibiotics usage and resistance among patients with COVID-19 in the intensive care unit (ICU) at the King Faisal Hospital in Makkah, Saudi Arabia in Makkah, Saudi Arabia.

I would like to hear more of 20,21 ref in the background- was it done in the ICU? If yes – what is new about your effort, or please specify the novelty of your research.

Concerning reference (20): "The COVID-19 pandemic: a threat to antimicrobial resistance containment". This study discusses the current knowledge on the SARS-CoV-2, and underscores the contribution of the COVID-19 pandemic on the escalation of AMR.

While, reference (21): "Increased antimicrobial resistance during the COVID-19 pandemic".

This study showed that there has been a rapid increase in multidrug-resistant organisms (MDROs), including extended-spectrum β-lactamase (ESBL)-producing Klebsiella pneumoniae, carbapenem-resistant New Delhi metallo-β-lactamase (NDM)-producing Enterobacterales, Acinetobacter baumannii, methicillin-resistant Staphylococcus aureus (MRSA), pan-echinocandin-resistant Candida glabrata and multi-triazole-resistant Aspergillus fumigatus. The cause is multifactorial and is particularly related to high rates of antimicrobial agent utilisation in COVID-19 patients with a relatively low rate of co- or secondary infection. 

None of the above-mentioned reference were conducted in the ICU. The novelty and the main strength of our study was it is the first study that shows the antibiotics usage and resistance among patients with COVID-19 in the ICU at the King Faisal Hospital in Makkah, Saudi Arabia.

If an English native speaker was not reviewing the text – please try to review phrases again: 

“Although the world is suffering to control the COVID-19 pandemic, the development of AMR outbreaks development should be considered.

We have corrected this sentence.

  1. METHODS 

2.1. Study design, setting, and period 

This cross-sectional study was conducted at the King Faisal Hospital in Makkah, Saudi Arabia, between November 1, 2020, and January 31, 2021.

Prospective? Retrospective? 

Prospective.

Only three months? How much perspective is there? 

The main limitations of this study are its cross-sectional design, which limits the generalizability of our results. Furthermore, as the study was cross-sectional, there was no follow-up for the study participants.

Hospital – how many beds?  In the hospital – how big is the population it serves, and how many icus? Who are the patients? Ards patients? What were the criteria for them as ards patients?  

Thank you for this important point. We have added the above suggested information.

Data collection: 

No need to remind the name and place of the hospital, city, or country so many times. 

Done.

Before describing the data collection, please specify – inclusion exclusions.

Done.

Primary and secondary outcomes

The primary outcome showed the prevalence of antibiotic resistance among ICU patients with COVID-19. The secondary outcome indicated that all bacteria were multidrug-resistance bacteria.

Samples from: body fluids? please specify- ascites is body fluid. Pleural effusions? Others? Nasal swaps? Swabs 

Pleural fluid, corrected.

Nasal swabs, corrected. 

Under sample size – “298 patients admitted to the ICU were eligible to be included in the study.”

This should be part of the results, not the methods. 

Thank you for this important point. We have transferred this sentence into the results section.

Only patients with positive COVID-19 (42 patients) were included in this study.--> what kind of patients? Intubated none? How sick? Under ab or not? Please describe the population – it is a highly heterogeneous one. 

In the current study, the inclusion criteria include all patients who were admitted to the ICU at the King Faisal Hospital during the study period, both genders, non-intubated spontaneously breathing patients, using antibiotics (narrow or broad spectrum) during the study period, all ages, with bacterial infection and who have a positive COVID-19 test were included in the study.

One sample was obtained from each patient according to his/her medical condition.--> please specify what it means according to a medical condition – definitions should be well described so that others can know who the studied population was and how to repeat your study somewhere else… 

In the current study, the results showed that 2 (4.8%) of the patients have acute myocardial infarction, 4 (9.5%) have acute pain, 1 (2.4%) have chronic kidney disease, 1 (2.4%) have dyspnea, 4 (9.5%) have heart attack, 4 (9.5%) have pneumonia, 3 (7.1%) have stroke, 13 (30.9%) have sepsis, 1 (2.4%) have weakness, 7 (16.7%) have unknown fever, and 2 (4.8%) have viral infection.

2.7. Statistical analysis 

The Statistical Package for Social Science (SPSS) version 24 (IBM Corp, Armonk, NY, USA) was used for data analysis, including descriptive statistics of frequencies and percentages.-

Not enough- what were the methods for analysis? Comparisons? Statistical tests? Etc.. 

Done.

  1. RESULTS 

A total of 42 patients with COVID-19 who were admitted to the ICU at the King Faisal Hospital in Makkah, Saudi Arabia, were included in the current study. Please explain what has happened from 298 eligible patients to only 42. 

Were these 42 positive for cultures? Unclear 

One sample from each patient? All 42 positive? 

In the currents study, out of 298 patients who were admitted to the ICU during the study period, only 42 patients with positive COVID-19 test were included in the final analysis. The inclusion criteria include all patients who were admitted to the ICU at the King Faisal Hospital during the study period, both genders, non-intubated spontaneously breathing patients, using antibiotics (narrow or broad spectrum) during the study period, all ages, with bacterial infection and who have a positive COVID-19 test were included in the study. The exclusion criteria include outpatients, patients with a negative COVID-19 test and patients with other types of microorganisms were excluded from the study. Additionally, one culture sample was obtained from each patient, and all of the obtained cultures were positive for bacteria.

Very small sample size – very hard to draw conclusions 

The study participants were selected using the census sampling method as all eligible patients were found during the study period (between November 1, 2020, and January 31, 2021). Actually, further future multicenter studies with large sample size, and long periods are recommended to confirm these findings.

The text under table 1 : 

Data are expressed as means ± SD for continuous variables and as a percentage for categorical variables. The differences between means were tested by using the independent sample t-test. The chi-square test was used to examine differences in the prevalence of different categorical variables. P value less than 0.05 was considered statistically significant. SD, stander deviation.

Should go to statistics in methods 

Done.

Discussion: 

“Despite the possibility of confounding factors, our investigation demonstrated that consistent with early findings [39], excessive antibiotic use was significantly related to a higher probability of MDRB acquisition in COVID-19 individuals. It is yet unclear if using broad-spectrum classes alone carries this risk or whether using any antibiotics at all.” There was no presentation regarding how and what antibiotics were provided to these patients before – so this conclusion is problematic

In the current study, all patients who were admitted to the ICU during the study period, both genders, non-intubated spontaneously breathing patients, using antibiotics (narrow- or broad- spectrum) during the study period, all ages, with bacterial infection and who have a positive COVID-19 test were included in the study.

  1. Conclusion 

In conclusion, our study demonstrates that antibiotic resistance is highly prevalent among ICU patients with COVID-19. Additionally, all of the obtained bacteria were multidrug-resistance bacteria.--> this is true in one specific hospital, not worldwide. 

The study conclusion was corrected.

Major: 

The sample size is a major factor here. Small numbers were collected in the very short time period in only one medical center.

We added to the study recommendation the following: Further future multicenter studies with large sample size, and long periods are recommended to confirm these findings.

The power of this study is very low as well as its generalizability 

not sure what conclusions can be drawn from it.

We added to the study limitations the following:

The main limitations of this study are its cross-sectional design, which limits the generalizability of our results. Furthermore, as the study was cross-sectional, there was no follow-up for the study participants. Additionally, all of the study isolates were collected from inpatients, and the exact number of nosocomial versus community-acquired bacteria was not differentiated. Further future multicenter studies with large sample size, and long periods are recommended to confirm these findings.

It is unclear what the ratio of positive cultures was – this 42 out of how many cultures? The study population is not well defined 

It is unclear if from 42 patients were taken only 42 cultures – one patient, one culture? If only patients that had growth of bacteria were included? How come we do not have more than one growth per patient? Other positive cultures were excluded. What was the median time in the ICU when these cultures were taken? What were the reasons for taking these cultures?

In the current study, all patients who were admitted to the ICU at the King Faisal Hospital during the study period, both genders, non-intubated spontaneously breathing patients, all ages, with bacterial infection and who have a positive COVID-19 test were included. One culture sample was obtained from each patient, and all of the obtained cultures were positive for bacteria. In addition, as the study was cross-sectional, there was no follow-up for the study participants, and it is one of the study limitations.

How many of them were on antibiotics? For how long? And how many? 

Were there any outbreaks of more in the ICU?

All patients were using antibiotics (narrow or broad spectrum) during the study period. We added to the inclusion criteria, in addition, as the study was cross-sectional, there was no follow-up for the study participants, and it was mentioned in the study limitations.

The authors argue that they have a high rate of MDR infections – but are not comparing to any other data – what were the rates of MDR infection for covid patients in different places around the world with different antibiotic abuse rates?

Done.

All these limitations -were not discussed.

Done. 

Round 2

Reviewer 2 Report

Second review:

Please double-check the references 17 and 22- how relevant they are to the sentence they refer to.

Under Methods:

This prospective cross-sectional study was conducted between November 1, 2020, and January 31, 2021 at the King Faisal Hospital in Makkah, Saudi Arabia, which includes 300 beds, serving 40k patients monthly more than 500k patients yearly, and the isolated ICU includes 30 beds.

Please edit the paragraph- 40K? or 40,000. Use Commas. Also- add the Population section here -where you describe the studied population – it is important to the flow of the reading.

The inclusion criteria include all patients who were admitted to the ICU at the King Faisal Hospital during the study period, both genders, non-intubated spontaneously breathing patients, using antibiotics (narrow or broad spectrum) during the study period, all ages, with bacterial infection and who have a positive COVID-19 test were included in the study. The exclusion criteria include outpatients, patients with a negative COVID-19 test and patients with other types of microorganisms were excluded from the study. Additionally, one culture sample was obtained from each patient, and all of the obtained cultures were positive for bacteria.

Why only – non intubated? Were they admitted due to COVID ARDS? Or any reason? Was it an isolated ICU for COVID patients? The term with bacterial infection should be changed to – with positive bacterial culture and??? How did you define bacterial infection ? fever? Other?

Results:

Concerning medical diagnosis of the patients, the results showed that 2 (4.8%) of the patients have an acute myocardial infarction, 4 (9.5%) have acute pain, 1 (2.4%) have chronic kidney disease, 1 (2.4%) have dyspnea, 4 (9.5%) have a heart attack, 4 (9.5%) have pneumonia, 3 (7.1%) have stroke, 13 (30.9%) have sepsis, 1 (2.4%) have weakness, 7 (16.7%) have unknown fever, and 2 (4.8%) have viral infection

have unknown fever.--> fever of unknown origin

have an acute myocardial infarction plus have a heart attackà isn’t it the same? heart attack is not a professional term

%) have acute painà is this a reason for icu admission?

Have pneumonia/strokeà suffered from instead of have?

have weaknessà is this a reason for icu admission?

I wonder – icu with covid patients – within the time frame of the surge of covid ARDS – where are these patients? Covid ARDS patients ? were they excluded? Please explain

Specify more in  Methods- was this a general ICU? Medical icu? Step down unit? How come none were intubated?

What about the SOFA score? Severity of illness-

Half of them died, and none were intubated? The studied population is not well defined – in the ICU literature this is the most important variable – what was the patient mix?

Discussion – start with your main findings and among who? I find The implanted paragraph does not fit there:

In the current study, all patients who were admitted to the ICU during the study period, both genders, non-intubated spontaneously breathing patients, using antibiotics (narrow or broad spectrum) during the study period, all ages, with bacterial infection and who have a positive COVID-19 test were included in the study.

MDRB spreads by cross-transmission or environmental factors in the hospital context, with pressure from antimicrobial therapy selection favoring certain individuals [23].- this sentence is not clear -what do the authors try to claim here? Please review

Due to the frequent and complicated caring, which makes it easier for the contamination of health-care personnel's hands and, as a result, the spread of MDRB [23], cross-transmission occurs in the ICU environment in between 23% and 53% of patient encounters [24,25]. Key elements in preventing the spread of MDRB are the implementation of infection prevention and control measures and the supervision of their observance [26].--> how does this sentence connect to your findings? – your cross-sectional study has no description of where from these infections were acquired? How long the patients were admitted pre to the positive cultures? Where all of these infections acquired in the hospital? Community acquired? à you have to mallow your conclusions – this was not part of the research design- you basically show a snapshot of a moment in time – not more than this- all other conclusions are not more than suggested causes with no statistical or methodological association.

Much of your discussion is based on the assumption that all positive cultures and ICU infections were acquired in the ICU- but there is no data to prove this -so the discussion and Conclusion are derived from a non- proved theoretical assumptions. Where is the data? For VAP / CLABSE and other ICU associated bacteremia and infections, there are specific definitions that are not shown here. You describe in the introduction that in your country (Saudi Arabia) many consume over the counter antibiotics – that may be a reason for all this MDR positive cultures – then in the discussion, you describe the reasons for acquiring MDR bacteria from the ICUà there is a gap here that is not connected in the design of this research. There was no genetic bacteria typing or any other investigation to show cross contamination in the unit .please rewrite discussion in a way that emphasizes that there is an observation (your results) and suggests explanations that were not proved in this study.

A firm conclusion cannot be formed on this topic due to the small number of patients who were worried;à what do you mean by worried?

Despite the possibility of confounding factors, our investigation demonstrated that, consistent with early findings [39], excessive antibiotic use was significantly related to a higher probability of MDRB acquisition in COVID-19 individuals.”--> how and where did you show this? This association was not shown in this research and cannot be made. Also, the referred study -39 – is not reoffering to your ICU or to the claim made… -all of these claims have to be mellowed dramatically.

The main strength of our study was it is the first study that shows the antibiotics usage and resistance among patients with COVID-19 in the ICU at the King Faisal Hospital in Makkah, Saudi Arabia.--> again the causality between antibiotics usage and the development of MDR is not shown here and the connection cannot be made upon the data presented

This limitation: Additionally, all of the study isolates were collected from inpatients, and the exact number of nosocomial versus community-acquired bacteria was not differentiated- to my view this is a major limitation that hampers most of the discussion.

The only main conclusion of this study is that MDR bacteria among ICU patients in this specific hospital , and time period is extremely high! Much higher than other reported ICU prevalence’s reported. The association with other factors- PPE, COVID, ICu behavior and others – are not shown and not associated. The reasons for their findings,  and the type of patients studied in this cohort are unclear. Time under AB exposure, time in the ICU pre-bacterial infection, all these are not explained. Basically there is not much more than a snapshot of cultures in a non-defined population and non-defined exposure …..

Despite the possibility of confounding factors, our investigation demonstrated that, consistent with early findings [39], excessive antibiotic use was significantly related to a higher probability of MDRB acquisition in COVID-19 individuals. It is yet unclear if using broad-spectrum classes alone carries this risk or whether using any antibiotics at all.à this is exactly the kind of linkage that is not shown or proved in the presented study nor explained in the article – the high MDR rate in the ICU among patients that are not classical COVID patients (no ARDS patients) is not associated to any of the explanation all are assumptions – we even do not know if these infections are hospital-acquired or community acquired.

Extensive English editing is needed

Author Response

Comments and Suggestions for Authors

Please double-check the references 17 and 22- how relevant they are to the sentence they refer to.

We have revised all references and the irrelevant references were omitted.

Under Methods:

“This prospective cross-sectional study was conducted between November 1, 2020, and January 31, 2021 at the King Faisal Hospital in Makkah, Saudi Arabia, which includes 300 beds, serving 40k patients monthly more than 500k patients yearly, and the isolated ICU includes 30 beds.”

Please edit the paragraph- 40K? or 40,000. Use Commas. Also- add the Population section here -where you describe the studied population – it is important to the flow of the reading.

Thank you for this important comment. We have corrected the values and added a new subsection called study setting.

 “The inclusion criteria include all patients who were admitted to the ICU at the King Faisal Hospital during the study period, both genders, non-intubated spontaneously breathing patients, using antibiotics (narrow or broad spectrum) during the study period, all ages, with bacterial infection and who have a positive COVID-19 test were included in the study. The exclusion criteria include outpatients, patients with a negative COVID-19 test and patients with other types of microorganisms were excluded from the study. Additionally, one culture sample was obtained from each patient, and all of the obtained cultures were positive for bacteria.”

Why only – non intubated? Were they admitted due to COVID ARDS? Or any reason? Was it an isolated ICU for COVID patients?

In the current study, we included all patients who were admitted to the ICU, non-intubated patients who complained of respiratory distress due to COVID-19 infection, and we excluded all intubated patients as there other serious causes for intubation and ICU admission. In addition, at the King Faisal Hospital there is a special an isolated ICU for COVID patients.

The term with bacterial infection should be changed to – with positive bacterial culture and???

Done.

How did you define bacterial infection? fever? Other?

In the current study, one culture sample was obtained from each patient, and all patients with positive bacterial cultures were defined as having bacterial infection; additionally all symptoms of bacterial infections were also assessed. 

Results:

Concerning medical diagnosis of the patients, the results showed that 2 (4.8%) of the patients have an acute myocardial infarction, 4 (9.5%) have acute pain, 1 (2.4%) have chronic kidney disease, 1 (2.4%) have dyspnea, 4 (9.5%) have a heart attack, 4 (9.5%) have pneumonia, 3 (7.1%) have stroke, 13 (30.9%) have sepsis, 1 (2.4%) have weakness, 7 (16.7%) have unknown fever, and 2 (4.8%) have viral infection

Have unknown fever.--> fever of unknown origin

Yes, modified.

Have an acute myocardial infarction plus have a heart attack isn’t it the same? Heart attack is not a professional term

In the current stud, 2 (4.8%) of the patients were diagnosed as having an acute attack of myocardial infarction. While 4 (9.5%) were diagnosed as having heart attack a condition marked by severe pain in the chest, often also spreading to the shoulders, arms, and neck, owing to an inadequate blood supply to the heart. Like in MI but, but in the ICU at the King Faisal Hospital the doctors were used this tem for patients with the same process of MI, which lasts long enough to cause permanent damage to the heart muscle.

Have acute pain is this a reason for ICU admission?

Have pneumonia/strokeà suffered from instead of have?

Have weaknessà is this a reason for icu admission?

In the current study, the main cause for ICU admission was respiratory distress due to COVID-19 infection not acute pain, pneumonia, stroke, weakness etc., but we assessed all of the main conditions and symptoms among the study participants.

I wonder – icu with covid patients – within the time frame of the surge of covid ARDS – where are these patients? Covid ARDS patients ? were they excluded? Please explain

Specify more in  Methods- was this a general ICU? Medical icu? Step down unit? How come none were intubated?

In the current study, the main cause for ICU admission was respiratory distress due to COVID-19 infection. In addition, at the King Faisal Hospital there is a special an isolated ICU for COVID patients.

What about the SOFA score? Severity of illness.

Unfortunately, the Sequential Organ Failure Assessment (SOFA) score was not used in the current study; we added to the study limitations.

Half of them died, and none were intubated? The studied population is not well defined – in the ICU literature, this is the most important variable – what was the patient mix?

In the current study, when we obtained the culture samples from the patients all of them were non-intubated, and we trying to avoid all intubated patients during data collection process to be more easy for us during data collection and we wrote that in that in the eligibility criteria. In addition, we know that there are other serious causes for ICU admission including the need for intubation and respiratory failure. Furthermore, as the study was cross-sectional, there was no follow-up for the study participants; just we reported the outcome of the participants without detailed investigations on the consequences.

Discussion – start with your main findings and among who? I find The implanted paragraph does not fit there:

In the current study, all patients who were admitted to the ICU during the study period, both genders, non-intubated spontaneously breathing patients, using antibiotics (narrow or broad spectrum) during the study period, all ages, with bacterial infection and who have a positive COVID-19 test were included in the study.

Done.

MDRB spreads by cross-transmission or environmental factors in the hospital context, with pressure from antimicrobial therapy selection favoring certain individuals [23].- this sentence is not clear -what do the authors try to claim here? Please review

Done.

Due to the frequent and complicated caring, which makes it easier for the contamination of health-care personnel's hands and, as a result, the spread of MDRB [23], cross-transmission occurs in the ICU environment in between 23% and 53% of patient encounters [24,25]. Key elements in preventing the spread of MDRB are the implementation of infection prevention and control measures and the supervision of their observance [26].--> how does this sentence connect to your findings? – your cross-sectional study has no description of where from these infections were acquired? How long the patients were admitted pre to the positive cultures? Where all of these infections acquired in the hospital? Community acquired? à you have to mallow your conclusions – this was not part of the research design- you basically show a snapshot of a moment in time – not more than this- all other conclusions are not more than suggested causes with no statistical or methodological association.

In the current study, we discussed the main possible causes for the spread of MDRB based on the results of the previous studies. Actually, further future multicenter studies with large sample size, and long periods are recommended to confirm these findings.

Much of your discussion is based on the assumption that all positive cultures and ICU infections were acquired in the ICU- but there is no data to prove this -so the discussion and Conclusion are derived from a non- proved theoretical assumptions. Where is the data? For VAP / CLABSE and other ICU associated bacteremia and infections, there are specific definitions that are not shown here. You describe in the introduction that in your country (Saudi Arabia) many consume over the counter antibiotics – that may be a reason for all this MDR positive cultures – then in the discussion, you describe the reasons for acquiring MDR bacteria from the ICUà there is a gap here that is not connected in the design of this research. There was no genetic bacteria typing or any other investigation to show cross contamination in the unit .please rewrite discussion in a way that emphasizes that there is an observation (your results) and suggests explanations that were not proved in this study.

The discussion section was modified.

In the current study, the main sources of positive bacteria culture and ICU infections were unknown, and further studies among ICU patients are recommended to determine the main sources of ICU infections such as (ventilator-associated pneumonia / central line-associated bloodstream infection, and other ICU associated bacteremia and infections).

 “A firm conclusion cannot be formed on this topic due to the small number of patients who were worried;”à what do you mean by worried?

In the current study, the study participants were selected using the census sampling method as all eligible patients were found during the study period (between November 1, 2020, and January 31, 2021). In addition, we added the following statement to the limitations of the study "Further future multicenter studies with large sample size, and long periods are recommended to confirm these findings".

“Despite the possibility of confounding factors, our investigation demonstrated that, consistent with early findings [39], excessive antibiotic use was significantly related to a higher probability of MDRB acquisition in COVID-19 individuals.”--> how and where did you show this? This association was not shown in this research and cannot be made. Also, the referred study -39 – is not reoffering to your ICU or to the claim made… -all of these claims have to be mellowed dramatically.

Modified.

The main strength of our study was it is the first study that shows the antibiotics usage and resistance among patients with COVID-19 in the ICU at the King Faisal Hospital in Makkah, Saudi Arabia.--> again the causality between antibiotics usage and the development of MDR is not shown here and the connection cannot be made upon the data presented

This limitation: Additionally, all of the study isolates were collected from inpatients, and the exact number of nosocomial versus community-acquired bacteria was not differentiated- to my view this is a major limitation that hampers most of the discussion.

We added the following sentences to the strength and limitations of the study: "the main limitations of this study are its cross-sectional design, the causal relationship could not be determined, and it limits the generalizability of our results. Additionally, further future multicenter studies with large sample size and long periods, and focusing on the main sources of ICU infections are recommended to confirm these findings".

The only main conclusion of this study is that MDR bacteria among ICU patients in this specific hospital , and time period is extremely high! Much higher than other reported ICU prevalence’s reported. The association with other factors- PPE, COVID, ICu behavior and others – are not shown and not associated. The reasons for their findings,  and the type of patients studied in this cohort are unclear. Time under AB exposure, time in the ICU pre-bacterial infection, all these are not explained. Basically there is not much more than a snapshot of cultures in a non-defined population and non-defined exposure …..

As the study has a cross sectional design we recommended further future multicenter studies with large sample size, long periods and follow up, and focusing on the main sources of ICU infections are recommended to confirm these findings. And we addressed that on the recommendations of the study.

Extensive English editing is needed

Done.

Round 3

Reviewer 2 Report

after all the efforts to improve the manuscript, I still think that the merit of the manuscript is not high enough. the population is not defined well, the generalizability is very limited, there is no association with any clinical or demographic data, and there is no significant learning point here. 

Author Response

Comments and Suggestions for Authors

after all the efforts to improve the manuscript, I still think that the merit of the manuscript is not high enough. The population is not defined well, the generalizability is very limited, there is no association with any clinical or demographic data, and there is no significant learning point here. 

Dear respected reviewer, please note that we have considered all of your valuable comments and other valuable comments by other reviewers as we can.  We appreciate the careful review and constructive suggestions. It is our belief that the manuscript is substantially improved after making the suggested edits.